# Leucine Reconstitutes Phagocytosis-Induced Cell Death in *E. coli*-Infected Neonatal Monocytes—Effects on Energy Metabolism and mTOR Signaling

**DOI:** 10.3390/ijms22084271

**Published:** 2021-04-20

**Authors:** Stephan Dreschers, Kim Ohl, Julia Möllmann, Klaus Tenbrock, Thorsten W. Orlikowsky

**Affiliations:** 1Section of Neonatology, University Children’s Hospital, Pauwelsstr. 30, 52074 Aachen, Germany; sdreschers@ukaachen.de; 2Department of Pediatrics, RWTH Aachen University, 52074 Aachen, Germany; kohl@ukaachen.de (K.O.); ktenbrock@ukaachen.de (K.T.); 3Department of Internal Medicine I, University Hospital RWTH Aachen, 52074 Aachen, Germany; jmoellmann@ukaachen.de

**Keywords:** monocyte, infection, apoptosis, neonatal sepsis, immuno-metabolism

## Abstract

MΦ differentiate from circulating monocytes (Mo). The reduced ability of neonatal Mo to undergo apoptosis after *E. coli* infection (phagocytosis-induced cell death (PICD)) could contribute to sustained inflammatory processes. The objective of our study was to investigate whether immune metabolism in Mo can be modified to gain access to pro-apoptotic signaling. To this end, we supplemented Mo from neonates and from adults with the branched amino acid leucine. In neonatal Mo, we observed increased energy production via oxidative phosphorylation (Oxphos) after *E. coli* infection via Seahorse assay. Leucine did not change phagocytic properties. In neonatal Mo, we detected temporal activation of the AKT and mTOR pathways, accompanied with subsequent activation of downstream targets S6 Kinase (S6K) and S6. FACS analyses showed that once mTOR activation was terminated, the level of anti-apoptotic BCL-2 family proteins (BCL-2; BCL-X_L_) decreased. Release of cytochrome C and cleavage of caspase-3 indicated involvement of the intrinsic apoptotic pathway. Concomitantly, the PICD of neonatal Mo was initiated, as detected by hypodiploid DNA. This process was sensitive to rapamycin and metformin, suggesting a functional link between AKT, mTOR and the control of intrinsic apoptotic signaling. These features were unique to neonatal Mo and could not be observed in adult Mo. Supplementation with leucine therefore could be beneficial to reduce sustained inflammation in septic neonates.

## 1. Introduction

Sepsis-induced mortality and morbidity peaks during infancy and although diagnostic and therapeutic efforts have been made, a recent statement of the WHO designates neonatal sepsis as a main threat for the future, since 9–20% of sepsis-suffering newborns still die [1]. Notably, preterm babies are vulnerable due to their immature organs and susceptible body and mucosal surfaces and become exposed to (hospital-specific) microbes, nutritional antigens, hyper- and hypoxia. The stressful transition from immune tolerance of the intrauterine surroundings to the situation after birth, with molecules initiating defensive immune responses, supports a state of sustained inflammation [2]. This may in turn cause or worsen sequelae such as necrotizing enterocolitis (NEC), retinopathy of prematurity (ROP), intracerebral hemorrhage (ICH), periventricular leukomalacia (PVL) and bronchopulmonary dysplasia (BPD) [3].

We previously demonstrated that monocyte-derived neonatal macrophages (MΦ) are defective in their metabolic repertoire, pointing to a potentially pivotal role of mTOR (mammalian target of rapamycin) signaling [4]. The mTORC1 complex measures the nutrient state of a cell and is involved in TLR-dependent, TRIF-mediated signaling, facilitating anti-inflammatory responses [5,6]. This signaling pathway is also thought to establish a tolerogenic state in alarmin (S100)-exposed neonatal Mo [7]. The mTOR protein can form two distinct protein complexes—mTORC1 and mTORC2. Whilst the function of mTORC1 requires the co-factor raptor, mTORC2 is formed by engaging rictor. The first protein complex is strongly inhibited by rapamycin, whereas the latter is frequently reported to be rapamycin resistant [8].

Additionally, mTORC1 can be stimulated upstream via IGF-receptor 1 and AKT signaling. Downstream, mTORC1 phosphorylates 4EBP and S6 kinase, thereby controlling the protein synthesis machinery. The activation of mTOR complex 1 (mTORC1) exerts anti-apoptotic reactions, but its reasonable to examine whether deactivation of mTOR shifts the balance towards pro-apoptotic pathways, e.g., by activating the intrinsic pathway.

Infections which lead to sepsis in adults are characterized by a loss of muscle mass and decreased de novo protein synthesis, primarily caused by a downregulation of mTORC1 activity [9].

The addition of branched-chain amino acids, among them the essential amino acid leucine, was not able to skew this situation. This defective response is termed “leucine resistance” [10,11] and little is known about whether leucine resistance also occurs in neonates [12]. In a neonatal pig model, malnutrition, low body weight and reduced muscle growth were correlated with obesity, neuronal deficits and cardiovascular diseases in later years [13]. The same study described that insulin and leucine rather than other branched amino acids such as valine and isoleucine increased protein and muscle synthesis [13,14]. Recent studies demonstrated that LAT-1, a transporter engaged in leucine import, modulates cytokine induced inflammatory reactions in T cells [15].

After infection with *E. coli* (the second most common species involved in neonatal sepsis), neonatal monocytes (CBMo) were shown to undergo phagocytosis-induced cell death (PICD) much less than Mo from adults (PBMo), due to alterations in the extrinsic, FAS- and TNF-dependent pathway as well as in the intrinsic pathway. Among a series of other factors, BCL-2 family proteins were shown to control a process, termed phagocytosis-induced cell death (PICD) [16,17,18,19]. Recently, we showed that the EGF-receptor ligand amphiregulin, which is upregulated in cord blood, can suppress PICD in cord blood-derived monocytes (CBMo) via upregulation of BCL-2 and BCL-X_L_. In turn, inhibition of the EGF-receptor increases PICD in *E. coli*-infected CBMo to levels of monocytes from adults (PBMo) [20]).

The PICD of effector cells, such as monocytes, may bias the immune response in either an inflammatory or anti-inflammatory manner by controlling the survival of certain subsets. Whereas direct bactericidal capacities of neonatal monocytes are conserved, they fail to secrete TNF-α as well as IL-10 and most importantly fail to induce a T-cell response and CD4^+^ T-cell expansion [19]. Monocytes which react inappropriately to pathogen exposure undergo PICD also via CD95/TNF-α mediated bystander apoptosis [16]. This phenomenon is reduced in neonatal monocytes and may contribute to a state of sustained inflammation with functional consequences, e.g., the elimination of other apoptotic debris and the persistent IL-8 secretion [21]. The identification of molecules involved in this crossroad of neonatal immunity could be beneficial for newborns suffering from sepsis and its sequelae.

We and others recently provided evidence for an energy failure caused by reduced glycolysis in neonatal monocytes [22] and MΦ [4]. Therefore, we addressed the question of whether the metabolic state of infected neonatal monocytes leucine supplementation could compensate the failure of energy metabolism in infected CBMo. To this end, we investigated the role of leucine supplementation and mTOR signaling in regard to bacterial phagocytosis. Furthermore, we wanted to clarify whether leucine has the potential to modulate PICD in CBMo.

## 2. Results

### 2.1. Glucose-Dependent Energy Metabolism Differs between PBMo and CBMo

Here, *E. coli*-infected PBMo increased glycolysis rates compared to CBMo (Figure 1A,C). The glycolytic capacity was also enhanced in naïve PBMo compared to naïve CBMo (Figure 1A, time interval from 21 to 30 min). It is noteworthy that infection provoked a significant difference in glycolysis between PBMo and CBMo (Figure 1A, compare 2nd and 4th columns).

The assessment of basal respiration showed comparable values for non-infected and infected PBMo (Figure 1B), whilst its values in non-infected and infected CBMo were significantly lower (Figure 1B). Taken together, the analysis revealed a reduced metabolic shift towards glycolysis in CBMo after infection. We considered whether leucine supplementation had an influence on energy metabolism after infection and found that leucine did not interfere with basal glycolysis in CBMo and that basal respiration was not increased significantly (Figure 1C). The addition of rapamycin to leucine-supplemented CBMo did not reverse the ECAR profile. Rapamycin enabled infected, leucine-supplemented CBMo to maintain a higher basal respiration when compared to infected CBMo (Figure 1D; *p* < 0.05)

Next, we assessed the expression of the leucine transporter LAT-1 (L-amino-acid transporter 1). This leucine transporter was two-fold stronger expressed in PBMo compared to CBMo (Figure 1E). Infection and the addition of leucine had no additional effect on CBMo (Figure 1F).

Finally, we assessed the glucose uptake and observed an equivalent uptake in uninfected PBMo and CBMo. In infected PBMo, the glucose uptake was double that of CBMo after an incubation period of 120 min (Figure 1G), suggesting that the observed effect could be caused by different glucose transporters which may lead to reduced glucose uptake.

### 2.2. Leucine Supplementation Does Not Interfere with Peri-Phagocytic Reactions

To exclude an influence of leucine on phagocytic properties, we analyzed the phagocytic index as well as phagocytic capacity of monocytes. Neither parameter was changed significantly by the addition of leucine (Figure 2A,B) and leucine treatment did not influence their ability to eradicate *E. coli* (Figure 2C).

### 2.3. Leucine Increases AKT and mTOR Activation in CBMo

AKT, which is a potential activator of mTORC1, was found to be more frequently activated in *E. coli*-infected PBMo than in CBMo (Figure 3C; columns 2 vs. 6). Infected CBMo did not activate AKT, but the addition of leucine increased AKT phosphorylation to levels of *E. coli*-infected PBMo (Figure 3C; columns 2 vs. 11). Interestingly, AKT phosphorylation of infected PBMo and infected, leucine-treated CBMo was found to be rapamycin sensitive (Figure 3C, compare hatched and blank columns). Although a high percentage of infected, leucine-treated CBMO were detected after 4 h p.i. (Figure 3C), at 24 h p.i. the p-AKT level of the same group was also elevated. However, the percentage of p-AKT expressing CBMo, treated with leucine or *E. coli*, respectively, was on the same level (Appendix A). In later time intervals (24 h p.i.), AKT phosphorylation was not rapamycin sensitive (Appendix A).

Accordingly, more infected PBMo activated mTOR via phosphorylation 4 h p.i. compared to the non-infected groups and to infected CBMo. The addition of leucine resulted in a significant activation of mTORC1 (Student’s *t*-test, *p* < 0.05, Figure 3D). Infected CBMo exhibited diminished phosphorylation. However, they reacted with a significant increase after the addition of leucine (Figure 3D, left panel). Rapamycin blocked mTOR phosphorylation in infected PBMo but not in leucine-treated, infected PBMo (compare hatched and blank columns in Figure 3D; *p* < 0.01). In infected CBMo, rapamycin blocked mTOR phosphorylation in both leucine-treated and non-treated, infected groups (compare hatched and blank columns in Figure 3D; *p* < 0.05).

We also tested mTOR phosphorylation after 24 h p.i. (Figure 3E). The phosphorylation pattern did not change for PBMo and CBMo reached a percentage of approximately 15% in all groups tested. To check whether phosphorylated mTOR is functionally active, we determined the phosphorylation of its target S6 ribosomal protein (Figure 3F) in the time intervals of 4 and 24 h p.i., respectively. PBMo did not show any alterations of S6 ribosomal protein phosphorylation after 4 h p.i. After 24 h p.i. S6 ribosomal protein phosphorylation was significantly upregulated in infected PBMo and leucine-supplemented, infected PBMo (*p* < 0.05 vs. non-treated). In CBMo, S6 ribosomal protein phosphorylation was increased in relation to basal levels. Similarly to PBMo no significant upregulation was observed 4 h p.i. However, 24 h p.i. *E. coli* and leucine-supplemented, *E. coli*-infected CBMo were enriched with phosphorylated S6 ribosomal protein. The results point to a delayed downstream activation of mTOR substrates in CBMo.

The phosphorylation of S6K (S6 Kinase), which is the direct mTORC1 downstream target in CBMo was not altered 4 h p.i. in all groups examined (Appendix A). Furthermore, we observed that additional leucine elevated S6K phosphorylation 24 h p.i. (Appendix A). The addition of leucine only caused an increase in phosphorylation. Thus, the elevated p-S6K level observed for *E. coli* infected, leucine-supplemented CBMo can be attributed to leucine. The S6K phosphorylation in the latter group was reversible by rapamycin, pointing to a mTORC1 dependency.

Next, we tested whether metformin could trigger the mTOR signaling in infected CBMo via an enhanced glucose uptake (Figure 3D, right panel). The addition of metformin did not increase mTOR activation (*p* < 0.5 vs. non-treated controls). In infected PBMo, metformin did not alter mTOR phosphorylation. However, the reduced mTOR phosphorylation in infected, leucine-supplemented PBMo could be skewed by metformin. Metformin did not change the proportion of infected CBMo expressing p-mTOR in both leucine-supplemented and control groups.

Although infection had a strong effect on 4EBP phosphorylation in PBMo, the addition of leucine did not increase the percentage of PBMo exhibiting p-4EBP. Leucine caused an increase in phosphorylation of infected CBMo within 4 h p.i., whereas infection alone had no effect (Figure 3G). After 24 h p.i, the number of PBMo exhibiting phosphorylated 4EBP decreased after infection, but could be restored after the addition of leucine (Figure 3G). Leucine treatment prior to *E. coli* challenge did not ameliorate 4EBP phosphorylation in CBMo 24 h p.i. (Figure 3G).

### 2.4. Leucine Increases PICD in CBMo in an mTOR-Dependent Manner

Since apoptosis contributes predominantly to the induction of PICD, we measured apoptosis rates in infected Mo which were treated with leucine. First, the induction of PICD between PBMo and CBMo (Figure 4A) revealed the diminished PICD of CBMo after *E. coli* infection, as described previously (16,18). However, incubation of CBMo with leucine enhanced PICD to levels observed for PBMo from 4 to 24 h. Supplementation of leucine to PBMo infected with *E. coli* did not cause an increase in PICD for 4 to 24 h.

Supplementation of the non-essential amino acid arginine prior to infection with *E. coli* enhanced PICD in CBMo (Figure 4B). This effect was significantly lower compared to supplementation with leucine before infection (Figure 4A).

The role of mTOR in the initiation of PICD was proven by the inhibitor rapamycin. Rapamycin not only reduced the PICD in infected groups, but skewed the leucine caused upregulation of PICD in infected CBMo (Figure 4, grey highlighted columns). The rapamycin dependency revealed that the mTORC1 complex is at least partially engaged.

Again, we checked the role of glucose by adding metformin to groups of Mo which were infected with or without leucine supplementation. Metformin, which sensitizes cells for insulin by increasing glucose uptake, did not increase PICD in infected PBMo. In contrast it reduced induction of PICD in infected PBMo treated with leucine as it could be shown for rapamycin. Additionally, metformin blocked the leucine-Induced PICD increase in infected CBMo (Figure 4, dark-grey highlighted columns).

Of note, only pretreatment with leucine enhances the PICD of CBMo. In experiments with consecutive addition of leucine, we did not observe any significant effect (Appendix A).

### 2.5. The Intrinsic Apoptotic Pathway is Engaged in the Leucine-Induced PICD

We next addressed which apoptotic pathway would contribute to the induction of leucine-triggered PICD.

As shown in previous studies, BCL-2 family proteins prevented PICD in CBMo [19]. Here, we compared the BCL-2 and BCL-X_L_ expression in *E. coli* infected, and leucine-supplemented, *E. coli*-infected groups (Figure 5A,B). After 24 h p.i. leucine pretreatment down-modulated BCL-X_L_ as well as BCL-2 levels in *E. coli*-infected CBMo, whereas *E. coli*-infected PBMo did not show any changes. Rapamycin blocked the infection-induced upregulation of BCL-2 and BCL-X_l_ (Figure 5A,B), suggesting that modulation of mTOR signaling may be engaged in this reaction. The higher disposability of agonists of the intrinsic apoptotic pathway caused an enhanced release of cytochrome C (Figure 5C) which then led to stronger induction of caspase-9 (Figure 5D). The expression of the pro-apoptotic factor PUMA was unaltered under every condition tested in PBMo and CBMo (Figure 5E).

We also studied the effect of leucine regarding components of the extrinsic apoptotic pathway (Figure 5F–H). Confirming earlier results, infection increased the TNF-α secretion in PBMo more than in CBMo (16). Leucine blocked the secretion in both PBMo and CBMo, but had no effect on the TNFR1 surface expression of CBMo. In addition, the expression of CD95L was not altered in leucine-pretreated, infected CBMo (Figure 5G). Induction of necrosis was found less prominent compared to apoptotic cell death (16).

## 3. Discussion

Here, we examined in vitro whether (i) *E. coli* infection and leucine supplementation alter the energy metabolism in CBMo and PBMo exceedingly and (ii) whether the signaling induced by leucine and infection affects the AKT/mTOR checkpoint with comparable reactions in CBMo and PBMo.

We observed a shift towards oxidative phosphorylation (Oxphos) in infected CBMo after leucine supplementation, whereas basal glycolysis was unaltered. This effect was correlated with an induction of PICD to levels detected for infected PBMo. The leucine-triggered metabolic response also correlated with increased phosphorylation of AKT and mTOR and, in turn, indirect downstream targets. Both infections induced metabolic response, and AKT/mTOR activation was terminated by leucine supplementation in addition to induction of the intrinsic apoptotic pathway in CBMo.

It has to be mentioned that two important signaling modules were already described to be missing/altered in CBMo. First, the extrinsic apoptotic pathway. CD95L/CD95- and TNFα/TNFR1-induced apoptosis is missing or reduced [16,18,23]. Accordingly, we did not observe adequate PICD in *E. coli*-infected CBMo (Figure 4A). Second, anti-apoptotic proteins are upregulated in CBMo compared to PBMo [19].

To summarize the results, we propose a a model for infected, leucine-treated CBMo addressing some of the role players (Figure 5I,J).

Since glucose uptake was found to be reduced (Figure 1G), which can be caused by the scarcity of glucose transporters (GLUT-1), leucine is metabolized to hydroxymethylbutyrate (HMB), which can fuel the acetyl-CoA pool [24]. HMB, an active intermediate of pyruvate, initiates acetylation of raptor and consecutively activates mTOR [25]).

This increases Oxphos, whereas ATP generation via the Warburg effect is infeasible (Figure 1A,B). Oxphos in turn strengthens mTORC1 phosphorylation. This effect is weaker, as detected in PBMo, which might be due to a reduced LAT transporter in CBMo (Figure 1E,F). Activation of mTORC-1 is boosted in leucine-supplemented, *E. coli*-treated CBMo (Figure 3D) compared to *E. coli*-treated CBMo. PBMo serve as a control and mimic the leucine effect by exhibiting an increased number of mTOR phosphorylated cells after adding leucine only, after *E. coli* infection and a combination of infection and leucine supplementation (Figure 3D). In infected CBMo, metformin is incapable to override leucine treatment (Figure 3D), which can be explained by a second signaling axis, driven by AKT. The AKT signaling after *E. coli* challenge was found unaltered in PBMo and CBMo after an interval of 24 h p.i. [20], which is in line with the results shown here (Appendix A). AKT can be activated by numerous stimuli among them growth factors and opsonized PAMPs via RTKs (receptor tyrosine kinases) and via TLR4/ERK signaling [26]. These stimuli first initiate PI3K (phosphatidylinositolkinase 3) upstream of AKT. Our results suggest that the PAMP/TLR4/PI3K signal is potentially transferred to AKT in PBMo, indicated by its phosphorylation. In the same interval, AKT activation requires leucine in CBMo (Figure 3A). Leucine could restore AKT and mTORC1 activation in infected CBMo to levels observed in infected PBMo, which received no leucine supplementation (Figure 3). It is interesting that both reactions can be reduced by rapamycin which maybe points to the involvement of mTORC2. Although mTORC2 was first described as rapamycin insensitive, a growing number of publications showed that mTORC2 is blocked by rapamycin depending on concentrations and length of incubation. Additionally, plausible is a regulating interaction between mTORC1 and mTORC2. Since we showed a high Oxphos, it is less likely that AMPK (AMP kinase) down-modulates mTORC1, but it was recently shown that AMPK upregulates mTORC2 activity via direct interaction with rictor [27]. One important function of mTORC2 is to control actin cytoskeleton rearrangement which is important to maintain phagocytosis. A LPS-dependent mTORC2 activation was shown to control Mo adherence [28]. Peri-phagocytic reactions were unaltered in CBMo compared to PBMo (Figure 2).

It is noteworthy that AKT activity continued for longer infection intervals in both CBMo and PBMo (Appendix A), but mTOR activity is more transient in infected CBMo irrespective of leucine treatment (compare Figure 3B,C). The mTOR signal is transferred to direct (4EBP-1, Figure 3G,H) and indirect (S6, ribosomal protein S6, Figure 3F) downstream targets. The results point to a faster reaction in PBMo and a slower, more unsteady reaction in CBMo (compare Figure 3F–H and Appendix A) in regard to the mean expression of phosphorylated targets as well as number of cells expressing the targets. One may speculate that the S6 kinase, which is the direct substrate of mTORC1, acts in a negative feedback mechanism terminating mTORC1 activity. As soon S6K down-modulates mTORC1, this anti-apoptotic balancer is blocked and anti-apoptotic factors are repressed. Our data are in line with this interpretation (Figure 5A–D). Although the expression of both BCL-X_l_ and BCL-2 can be blocked by rapamycin and is leucine dependent, further studies have to prove whether these effects are causatively linked or coincidental. The signaling of AKT and mTOR may act independently. However, it seems to be obligatory to prime CBMo with leucine before infection to obtain a PICD (Appendix A). The role of the AKT and mTOR signaling and the regulation of BCL-2 family proteins was investigated in leukemic U937 cells. In this study, the inhibition of mTOR and the ERK pathway initiated apoptosis [29].

Most of the previous studies focusing on leucine supplementation- and infection-induced effects are based on cell culture or address effects in muscle and other organs. A comparison of neonatal monocytes subjected to two different stimuli (*E. coli* and leucine) is a complex experimental setup with results that have to be interpreted carefully due to experimental limitations.

One of the limitations is the dependency on inhibitor substances such as rapamycin. As pointed out before, rapamycin can exert various effects which are dependent on concentration and exposure time. In a study comparing the effects of rapamycin on dendritic cells and Mo, no autonomous initiation of apoptosis in Mo and MΦ was reported [30]. This is in line with our results, since the administration of rapamycin alone did not change the induction of apoptosis in CBMo and PBMo.

Additionally, rapamycin was shown to have T cell-specific functions—it suppresses mTORC1, but activates mTORC2 signaling [31].

The role of metformin has to be clarified. Additionally, it is reported to interfere with mTOR activation [32]—it is a janiform drug, which increases glucose uptake and keeps mTORC1 deactivated (see Figure 5I). This could lead to opposite reactions. Indeed, in PBMo, we observed no effect for *E. coli*-treated Mo and a tendency to further increase mTOR activation of leucine-pretreated, infected PBMo. Regarding PICD, metformin caused anti-apoptotic effects predicted by AKT/mTOR signaling by returning apoptosis of leucine-supplemented, infected PBMo to values of non-infected PBMo. In CBMo, metformin had the same effect, but had no detectable effect on mTOR signaling. This might be due to inappropriate synchronization of the different stimuli elected. Further experiments are planned to dissect this signal transduction pathway more concisely.

As shown for senescent cells [33], PAMP-challenged neonatal monocytes are characterized by increased metabolic demands and are vulnerable in regard to a less effective glucose metabolism. We observed decreased glucose uptake in *E. coli*-treated CBMo (Figure 1G) and added metformin to compensate a potential shortness of cellular glucose levels. Metformin has also been reported to interfere with mTOR activation [32]. In our experiments, metformin failed to increase PICD rates after infection in CBMo. (Figure 4), suggesting that glycolysis is not directly linked to pro-apoptotic mechanisms in CBMo. Furthermore, mTOR was not shown to be more strongly activated by metformin in *E. coli*-infected PBMo and CBMo (Figure 3D).

Upon challenge with PAMPs such as LPS, MΦ exhibit a Warburg effect characterized by a metabolic shift from oxidative phosphorylation to glycolysis [34]. The regulation of monocytic glycolysis upon *E. coli*/LPS activation is not completely understood [35,36]. Infection-triggered upregulation of glycolysis has been reported [35], suggesting a dependency on LPS concentrations and distinct PAMPs. LPS did not increase basal glycolysis, but *E. coli* lysates did increase glycolysis two-fold [37]. The same study described an increase in basal respiration after *E. coli* challenge, which is not in line with our observations for PBMo (Figure 1B). Publications which compare Oxphos profiles of neonatal and adult leucocytes are scarce. Here, the basal respiration of non-treated CBMo was lower compared to PBMo and a comparable profile was found for ATP-linked respiration (Figure 1B).

On the other hand, the PICD initiation by leucine in CBMo is quite specific and not attributed to a general function of amino acids, since arginine increased PICD in infected CBMo to a two-fold lesser extend (Figure 4B).

Numerous studies indicate that supplemental nutrition can modify cellular energy metabolism which regulates immune responses [38,39,40] and that supplementary nutrition is beneficial for patients suffering from severe infections and sepsis [41,42]. Supplementation with high concentrations of protein, L-carnitine and lactoferrin as well as immunoglobulins [43] have been tested. When a high-protein diet was combined with ß-hydroxy-ß-methyl-butyrat (HMB), patients recovered more efficiently from infections [44].

The activation of mTOR is published to be mainly anti-apoptotic. A pro-apoptotic effect after mTOR deactivation is described for B cells [45]. Moreover, the induction of necrosis in CD4^−^/8^−^ T cells of *Lupus erythematosus* patients was reported to be mTOR dependent [46]. Worth mentioning, the detection of intrinsic apoptotic markers is limited [47]. The decrease in anti-apoptotic BCL-2 like proteins such as BCL-X_L_ which we observed after leucine pretreatment could therefore be explained by a bypassing of the glucose fueled acetyl-CoA pool. As a consequence, cytochrome C efflux initiated by phagocytic signals cannot be blocked sufficiently and cleavage of caspase 9 and 3 initiates apoptosis (Figure 5A–D). Inhibition of glucose metabolism was shown to reduce an AKT/mTOR-dependent upregulation of MCL-1, another protein of the BCL-2 family, in cultured B cells [45]. After blockade of glycolysis, the B-cell line underwent apoptosis. The B-cell study also provided evidence that blockade of glycolysis downregulates MCL-1. Here, lower expressions of BCL-X_L_ and BCL-2 in leucine-treated CBMo could be detected at later time points (24 h p.i.). Comparing S6 ribosomal protein activation downstream from mTOR in leucine-treated, infected CBMo revealed deactivation of S6 ribosomal protein, which could result in a downturned de novo synthesis of anti-apoptotic proteins (Figure 3F and Figure 5A,B), whereas the level of the pro-apoptotic factor PUMA was unaltered (Figure 5E). Further studies are planned to dissect this mechanism in monocytes. Furthermore, we could exclude a role of the extrinsic apoptotic pathway (Figure 5E–G), which is a key signaling pathway in PICD, supporting bystander apoptosis [48].

We found a role of mTORC1 in leucine-mediated PICD in infected CBMo. The mTORC1 complex has been shown to be involved in anti-apoptotic pathways. In our experimental setup, mTORC1 is activated by leucine, but this activation is terminated after 24 h (Figure 3D). Additionally, downstream target S6 activation is terminated 24 h after the addition of leucine at an intracellular level, which is lower than for untreated cells (Figure 3F). The decrease in anti-apoptotic signaling via AKT/mTORC1 in combination with the decreased concentrations of BCL-2 and BCL-XL (Figure 5A,B) could contribute to the increased PICD in the leucine-treated, infected monocytes.

To confer the non-specific, innate immune response protective against second infections, monocytes could be pre-exposed to PAMPs. These experimental setups are termed “trained immunity”, which becomes an attractive therapeutic issue [37] to induce T-cell activity [48]. LPS-challenged MΦ with a leucine substrate analogue (ERG240) reduced glycolysis and established an anti-inflammatory reaction [36]. Furthermore, studies reported mTOR-regulated metabolism as a therapeutic target to heal autoimmune diseases (systemic lupus erythematosus (SLE)) [46]. This study demonstrated that regulation of metabolic reactions via leucine stimulation could anticipate a pathogenic attack. Enhanced PICD in neonatal Mo could lower hyper- as well as sustained inflammatory reactions, which may be beneficial for the newborn. For leucine, which represents an easy to administer therapeutic agent, an extended in vivo analysis of its role in immune modulation with regard to neonatal sepsis could be a future tool.

## 4. Material and Methods

### 4.1. Patients

The study protocol was approved by the Ethics Committees of Aachen University Hospital (Permission No. EK150/09, 6 October 2009, signed by Profs G. Schmalzing and U. Buell). All adult participants involved were informed and gave written consent to use their blood samples for this study. All term neonates were delivered spontaneously and did not exhibit signs of infection, as defined by clinical status, white blood cell count and C-reactive protein. Mothers with amnion infection and prolonged labor (>12 h) were excluded. Umbilical cord blood was placed in heparin-coated tubes (4 IE/mL blood), immediately following cord ligation, as described before [43]. All methods were performed in accordance with the relevant guidelines and regulations.

### 4.2. Reagents

Primary antibodies and Ig-matched controls (IgG1, IgG2b) were from BD Biosciences (Heidelberg, Germany), Thermo Fisher Scientific (Frankfurt, Germany), Santa Cruz Biotechnologies (Heidelberg, Germany) and Immunotools (Friesoythe, Germany). Isopropyl-β-D-thiogalactopyranoside (IPTG) and antibiotics were purchased from Sigma (Munich, Germany). The glucose uptake assay kit was also provided by Sigma. Leucine and arginine were purchased from Roth (Karlsuhe, Germany). Rapamycin, metformin and non-opsonized *E. coli* pHrodo particles were from Thermo Fisher Scientific. The antibodies to CD14 (clone MEM-18), phospho-mTOR (serine 2448, clone MRRBY), phospho-AKT (phospho serine 473 AKT1, 2, 3, rabbit polyclonal AF887), phospho-S6-Kinase (p-S6K, phospho-Thr 421, phosphor-Ser 424, clone E-5), phospho-S6 ribosomal protein (serine 235, serine 236, clone N7-548), PUMA (goat polyclonal IgG), LAT-1 (rabbit polyclonal IgG) phospho-4EBP (threonine 36, threonine 45, clone M31-16) BCL-X_L_ (clone 7B2.5), BCL-2 (C-2), cleaved caspase-9 (clone D819E) and cytochrome C (clone 7H8.2C12) were used in dilutions recommended by the supplier.

### 4.3. Isolation of Mo and Stimulation

In brief, leukocytes were prepared by density gradient centrifugation of whole blood from healthy adult donors and cord blood from term neonates, as described before [43].

The phagocytosis index (CD14^+^GFP^+^ MΦ %: CD14^+^ MΦ %) and the phagocytic capacity (mean fluorescent intensity (MFI) of CD14^+^ monocytes) were assessed by flow cytometry after 4 h p.i. (post-infection) and 24 h p.i. For bacterial phagocytosis assays, a multiplicity of infection of 50 (MOI 50) was utilized.

The bacterial killing assay was performed as described elsewhere [15]. In brief, *E. coli* were allowed to infect Mo for 1 h at standard conditions. Afterwards, the medium was replaced by medium containing antibiotics for additional 3 h. Mo were washed in PBS prior to disruption by the addition of 0.1% *v*/*v* Triton-X 100 and vigorous shaking. After centrifugation, the pellet was resuspended in LB and applied to LB agar plates.

We added leucine in a concentration range from 0.5 to 50 µg/mL prior to and consecutively with *E. coli* to CBMo (Appendix A). Leucine supplementation increased PICD significantly only after pretreatment in a concentration range from 5–50 µg/mL leucine. The addition of leucine only had no effect. We decided to conduct all further experiments with 5 µg/mL leucine 30 min prior to infection. Rapamycin was added to a final concentration of 50 nM, 30 min before infection with or without leucine. Metformin was added to a final concentration of 2 mmol/L 30 min prior to infection.

### 4.4. Flow Cytometry

A daily calibrated FACS-Canto flow cytometer (Becton Dickinson, MountainView, CA, USA) was used to perform phenotypic analysis. To prevent non-specific binding, cells were incubated with 10% fetal calf serum on ice for 10 min before staining with pacific-blue (PB)-, fluorescein-isothiocyanate (FITC)-, phycoerythrin (PE)-, allophycocyanin (APC)-, or anti-IgG secondary-labelled monoclonal antibodies for 45 min in the dark. Monocytes were gated using forward scatter (FSC), side scatter (SSC), and CD14 expression.

Cells were fixed in 1% paraformaldehyde for 1 h and permeabilized with 0.1% *v*/*v* Triton-X100 in PBS for 5 min. Afterwards cells were washed and blocked with PBS/FCS (5% *v*/*v*) for 20 min at room temperature (RT). Cells were washed again and stained with fluorochrome-labelled antibodies in PBS/FCS (5% *v*/*v*) for 60 min at RT followed by additional washing. Intracellular staining of phosphorylated mTOR, S6 ribosomal protein and AKT were conducted following the manufacturer’s recommendations. Flow cytometric results were expressed as a percentage of positive cells. In some charts, the mean fluorescent intensity (MFI) is presented. For the phospho-AKT staining, the results were checked by Western Blot analysis (see Section 4.7. and Appendix A).

### 4.5. Seahorse Assay

In total, 2 × 105 cells were seeded on gelatin-coated plates and OCR/ECAR was measured using the XF96 Extracellular Flux Analyzer (Seahorse Bioscience, Billarica, MA, USA) following the manufacturer’s instructions. OCR was measured in XF media containing 11 mmol/L glucose and 1 mmol/L sodium pyruvate under basal conditions and in response to 1 μmol/L oligomycin, 1 μmol/L carbonyl cyanide p-trifluoromethoxyphenylhydrazone (FCCP), and 0.1 μmol/L rotenone plus 0.1 μmol/L antimycin A. The extracellular acidification rate (ECAR) was measured in assay medium (DMEM supplemented with 4.5 g/L glucose and 2 mM glutamine) under basal conditions and in response to 10 mM glucose, 1 M oligomycin, and 100 mM 2-deoxyglucose. Each value represents the mean of pentaplicates of three independent experiments. Basal respiration/glycolysis rates were determined by calculating the mean of three time points indicated by grey areas in Figure 2 and Figure 3. The ATP production was calculated by subtracting the mean proton leak (mean of time interval 12–18 min) of an OCR profile from the mean of basal respiration.

### 4.6. ELISA

The TNF enzyme-linked immunosorbent assay (ELISA) was purchased from eBiosciences (eBiosciences-Natutec, Frankfurt, Germany). It was used according to the manufacturer’s recommendations. The read out was executed in a spectra max 340PC ELISA reader (Molecular Devices, Sunnyvale, CA, USA) with a sensitivity of 4–500 pg/mL.

### 4.7. Western Blot

For validation of FACS results (Appendix A), Ficoll-purified Mo were plated on cell culture dishes. After 30 min standard cultivation, the supernatant was removed and the resting cell population identified as Mo enriched. This Mo population was subjected to the protocols described above and later used for Western blot experiments. Isolation of the cytosolic fraction of Mo was performed by resuspending 5 × 10^6^ treated and non-treated monocytes with 300 µL PBS containing a protease inhibitor cocktail (Carl Roth GmbH, Karlsruhe, Germany) and subjecting the cells to three rounds of freeze–thaw cycles. Afterwards, the crude cell lysate was mixed with the 4 times concentrated Lämmli buffer. After boiling for three minutes, the lysates were deployed to SDS-PAGE. Proteins from the SDS PAGE were transferred to nitrocellulose membranes using the semi-dry blotting technique (according to standard protocols of Lämmli and Khyse Anderson). Non-specific binding was blocked by incubation in 5% *w*/*v* milk-powder in PBS for 2 h at RT. The cytochrome C antibody was diluted 1:1000 in TBST (Standard TBS, supplemented with 0.1% Tween *v*/*v*) and incubated overnight at 4 °C. After 4 washes in TBST the secondary antibody was diluted 1:5000 in TBST. After 1 h incubation at RT, the membrane was again washed twice and signals were detected utilizing the kit. For imaging and quantification, a LAS 3000 imager (Fujifilm, Düsseldorf, Germany) combined with the Multi-Gauge software (Fujifilm, Düsseldorf, Germany) was used.

### 4.8. Statistical Analysis

In all experiments, results are expressed as the mean +/− standard deviation. Error bars represent standard deviations. Values of *p* < 0.05 were considered significant. Analyses were performed with statistical software performing Student’s *t*-test and two-way ANOVA. Experiments where N = 3 were tested according to the Mann–Whitney test for significant differences. Data which did not pass a test for Gaussian distribution were tested with a Kolmogorov–Smirnov test, as provided by Graph Prism Pad Software Statistical Package, La Jolla, CA 92037 USA.

## Figures and Tables

**Figure 1 ijms-22-04271-f001:**
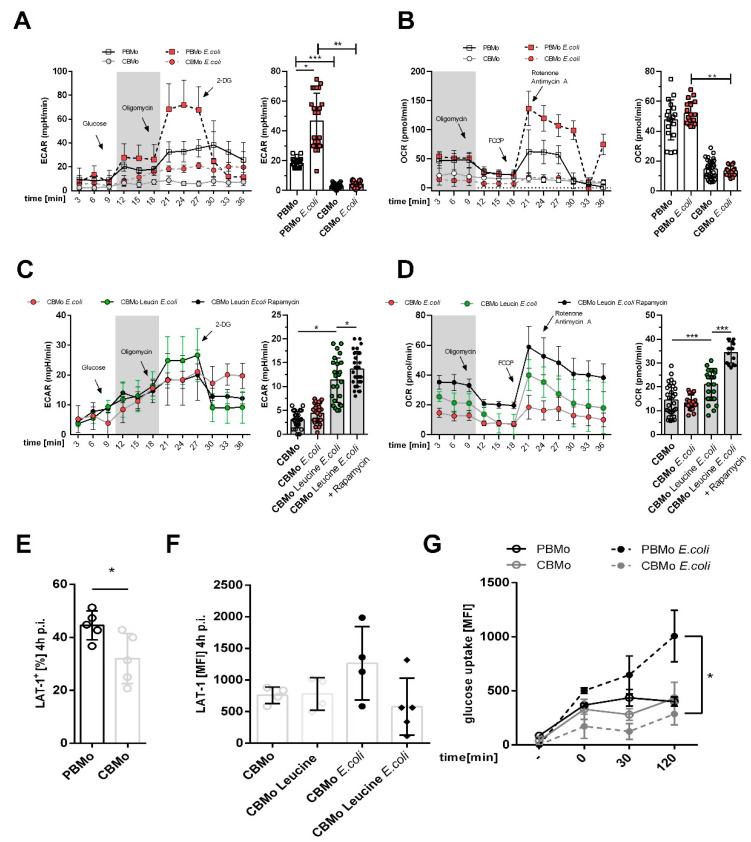
Energy metabolism in Mo. Extracellular acidification rate (ECAR) profile of indicated groups ((**A**), error bars give standard deviation (SD)). The comparative summary of basal glycolysis (grey area in (**A**)) is given in the chart to the right. OCR (oxygen consumption rate) profile of indicated groups ((**B**), error bars give SD). The chart to the right summarizes the basal respiration (grey area in (**B**); bars, Student’s-*t*-test, * *p* < 0.05; blunt ended bars, ANOVA analysis; ** *p* < 0.01; *** *p* < 0.005). ECAR chart ((**C**), error bars give SD) and OCR chart ((**D**), error bars give SD) compares infected CBMo. Indicated groups received leucine and rapamycin, respectively. The comparative summary of basal glycolysis (grey area in (**C**)) and basal respiration (grey area in (**D**)) is given in the charts to the right (error bars represent SD; Student’s *t*-test, * *p* < 0.05; *** *p* < 0.005). LAT-1 (L-amino-acid transporter 1) expression was assessed in non-treated PBMo and CBMo ((**E**); ANOVA analysis; ** *p* < 0.01) and in CBMo after infection and leucine treatment (**F**). The glucose uptake of indicated groups was assessed within the given time interval via fluorometric detection of metabolized 2-deoxyglucose ((**G**); error bars represent SD; Student’s *t*-test; * *p* < 0.05).

**Figure 2 ijms-22-04271-f002:**
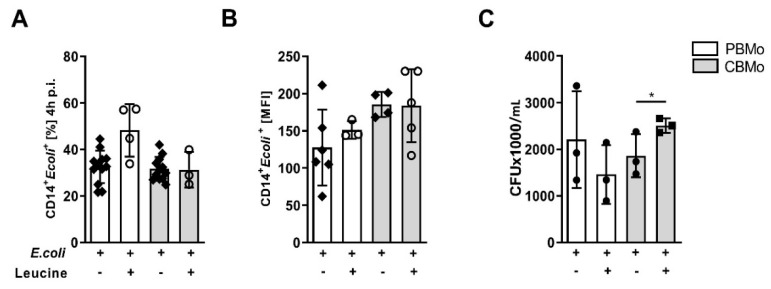
Peri-phagocytic responses to infection and leucine supplementation. Phagocytic index (**A**), phagocytic capacity (**B**) (multiplicity of infection 50) and bacterial survival ((**C**); Student’s *t*-test * *p* < 0.05) were assessed. All error bars indicate standard deviation (SD).

**Figure 3 ijms-22-04271-f003:**
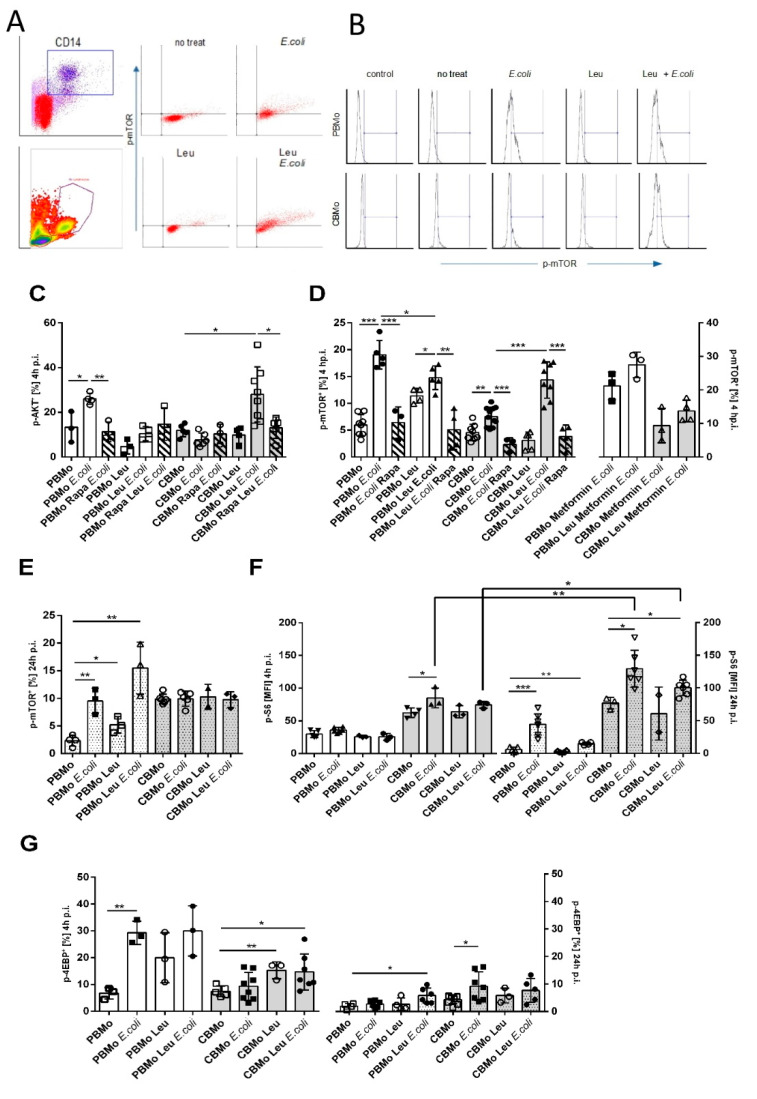
Leucine interferes with AKT/mTOR signaling. Gating strategy is depicted in (**A**). CD14-positive cells (dot plot) were back-gated to confirm Mo characteristics via position (density plot). Dot plots to the right show PBMo exhibiting phosphorylated mTOR. Histogram analysis (**B**) of the FACS-based p-mTOR staining for PBMo and CBMo under indicated conditions. The percentage of Mo exhibiting phosphorylated AKT was determined (**C**). The percentage of Mo expressing the phosphorylated form of mTOR was assessed (**D**). Indicated groups received leucine and metformin, respectively. Hatched columns indicate groups which received rapamycin (Rapa). Please note the different y-axis scale in the left and right panel of (**D**). (**E**) shows phosphorylated mTOR for a 24 h p.i. infection interval and (**F**) the phosphorylation of mTOR substrate S6 ribosomal protein in infection intervals of 4 h.p.i. (blank columns) and 24 h p.i. (dotted columns). The percentage of Mo exhibiting the phosphorylated mTOR substrate 4EBP is given in (**G**) for the indicated infection interval (all charts give error bars representing the standard deviation (SD); Student’s *t*-test * *p* < 0.05; ** *p* < 0.01, *** *p* < 0.005).

**Figure 4 ijms-22-04271-f004:**
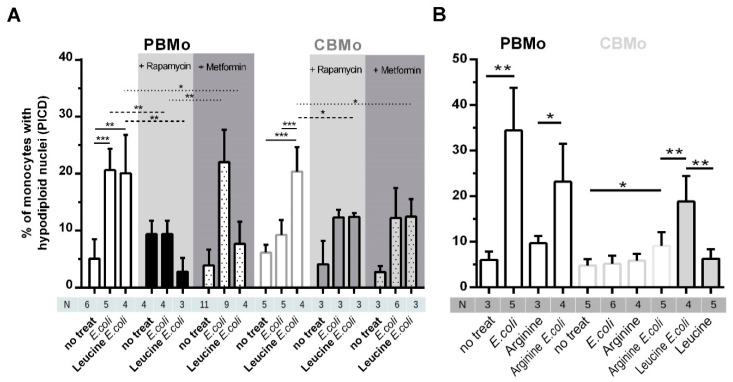
Leucine-Induced PICD is dependent on mTOR signaling. PICD was compared between PBMo and CBMo for the infection interval of 24 h. The multiplicity of infection was 50. Indicated groups received rapamycin (50 nM, 30 min before infection grey highlighted) or metformin (2 mmol/L 30 min before infection, dark-grey highlighted; Student’s *t*-test * *p* < 0.05; ** *p* < 0.01; *** *p* < 0.005) Indicated groups were cultured with leucine (5 µg/mL, 30 min before infection). (**B**) Comparable experimental setup as shown in (**A**). Mo were pre-incubated with l-arginine (6.5 µg/mL) and indicated groups infected with *E. coli*. Apoptosis was detected 24 h p.i. (Student’s *t*-test * *p* < 0.05). All charts give error bars representing the standard deviation (SD).

**Figure 5 ijms-22-04271-f005:**
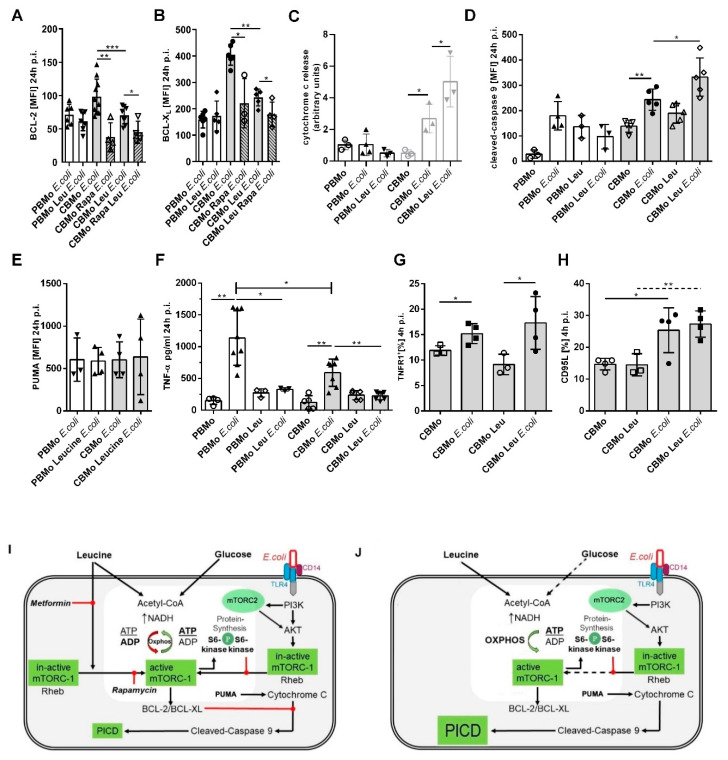
Leucine-dependent PICD activates components of the intrinsic apoptotic pathway. The median expression of BCL-2 (**A**) and BCL-XL (**B**) was assessed under indicated conditions. Rapamycin (Rapa) was added together with leucine (Leu). Cytochrome C release was detected via immunoblot analysis. Cytosolic cytochrome c was quantified from the independent number of immunoblots indicated (**C**). Caspase-9 cleavage (**D**) and the pro-apoptotic factor PUMA (**E**) was compared between PBMo and CBMo for the infection interval indicated (Student’s *t*-test * *p* < 0.05; ** *p* < 0.01; *** *p* < 0.005, ANOVA test in (**F**), * *p* < 0.05). (**G**) CBMo expressing TNFR1 and CD95L (**H**) were assessed via FACS staining (Student’s *t*-test * *p* < 0.05; ** *p* < 0.01). MOI 50 for all experiments. Error bars in all charts represent standard deviation (SD). The drawing (**I**) summarizes the cross-talking signal transduction pathways. RTK, receptor tyrosine kinases; mTOR, mammalian target of rapamycin; BCL-2, BCL-X_L,_ PUMA, apoptosis modulators belonging to the B-cell lymphoma 2 protein family; CD14, LPS receptor; TLR4, Toll-like receptor 4; Oxphos, oxidative phosphorylation; Rheb, Ras homologue enriched in brain. The reactions for infected, leucine-treated CBMo are given in the drawing beside (**J**).

## Data Availability

Original data can be found at: 10.6084/m9.figshare.14446992.

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
