# Peer review of "Leucine Reconstitutes Phagocytosis-Induced Cell Death in E. coli-Infected Neonatal Monocytes—Effects on Energy Metabolism and mTOR Signaling"

_ijms, 2021, doi:10.3390/ijms22084271_

Round 1

Reviewer 1 Report

Overall revised results are acceptable

Reviewer 2 Report

In this revised form of the manuscript, many of the comments made by the referees have been taken into consideration. However, the revision seems to be rather superficial on some points. The internal quality check and proof reading have not been highly prioritized, i.e. abbreviations are not explained the first time they are used, the figure legends are still incomplete, incomplete sentences, a mix of lowercase and uppercase, and a lot of punctuation mistakes.

Concerning the methodology, information on how the glucose uptake was performed is not included. The usefulness of metformin in studying the role of glucose uptake can also be questioned. 

At the beginning of the results, some previous results are presented apparently without much relevance (line 112-113). I suggest that this sentence is removed. The figures could also be more consistent, i.e. Fig.3 F contains p6S after both 4h and 24h, whereas p4EBP is divided into Fig.3 G 4h and Fig.3 H 24h.

The accessibility of the paper, and the discussion in particular, can be improved. Highlights of the findings should be made before the details are discussed in relation to findings by others. The "story",  with its interpretations and implications, is still quite hard to follow.

The pdf-file with specific comments is attached, notice that just some examples are marked.  A final proof-reading must be performed.

Reviewer 3 Report

1) Authors have to provide the phospho epitopes that were analysed otherwise it is not possibleto correctly analyse the results: for instance was it pAKT ser 473 or thr308?

Thank you for this suggestion. We used a phospho AKT antibody, which detects ser 473phosphorylation site. This now has been added to the Material & Methods section (ll 462-465). The phosphorylation status is reflecting stress levels (for example lack of glucose, doi:10.18632/oncoscience.16. eCollection 2014) it would have been interesting to study a differencebetween single and double phosphorylation, since AKT ser 473 phosphorylation ismaintained by mTORC2 and enhances AKT activity. However, as you pointed out, the temporalactivation of AKT can be caused by different stimuli (for example phagocytosis).

The studied phospho epitopes of mTOR, 4E-BP1 and S6 ribosomal protein are still missing

2) Leucine is a major activator of mTORC1 however the authors failed to demonstratemTORC1 activation following leucine treatment. S6 ribosomal protein is not a direct downstream effector of mTORC1 and gets activated only after 24h. In addition there were noincrease in 4EBP-1 phosphorylation.

In Figure 3 d we showed a significant phosphorylation of mTORC1 accompanied by downstream

phosphorylation of S6 ribosomal protein for stimulation with E.coli and leucine (Column

6 vs 8). mTORC1 is phosphorylated by leucine only (Student`s t-test p<0.001). We have

added the sentence: “Addition of leucine resulted in a significant activation of mTORC1.” (ll

175-176)

The fact, that 4EBP does not react similarly to S6 does not necessarily mean that mTORC1

is not functional. The review of D. Sabatini shows an activation of S6 ribosomal protein and

inhibition of 4EBP-1 via phosphorylation (citation doi: 10.1038/s41580-019-0199-y Fig.2a /

Fig.5).

Following activation mTORC1 phosphorylates 4E-BP1 and S6K1. In turn S6K1 phosphorylates S6 ribosomal protein. Here, leucine induces mTORC1 phosphorylation in infected CBMO after 4hours (Fig3D) but no increase of 4E-BP1 phosphorylation nor S6 ribosomal protein are detected at 4 hours (Figure 3F and G). Hence, mTORC1 activity is not increased by leucine in infected CBMO unless the authors demonstrate clear increased phosphorylation of another mTORC1 downstream effector at 4 hours.

3) Authors state that mTORC1 has proapoptotic effects and cite a 2002 reference (number 10 ).This is not adequate as in 2002 mTORC2 was not even discovered. In fact, very few evidencehave demonstrated that mTORC1 induces apoptosis.

Thank you for this comment. We started with the review of Sabatini, followed by citations ofPerl et al., therefore have changed the appearance of citations. In order to give respect tothis point, we now have changed the sentence in the introduction to ”The activation of mTORcomplex 1 (mTORC1) exerts anti-apoptotic reactions, but additional studies point towardspro-apoptotic processes by deactivation of mTOR and anti-apoptotic BCL2-family proteins,shifting the balance towards pro-apoptotic pathways, e.g. by activating the intrinsic pathway(10).”

Please provide strong recent evidences from the litterature that in 2020 mTORC1 exhibit pro-apoptotic effects. The idea in 2002 that mTORC1 might regulate Bcl2 and bclxl was not confirmed later.

4) Leucine increases pAKT in CBMo infected cells. What is the mechanism behind this effect.Also AKT is a major anti apoptotic signaling intermediary which fully contrasts with the results obtained by the authors.

We tested AKT phosphorylation in intervals of 4 h and 24 h p.i.. As you pointed out before,AKT got phosphorylated in combination with infection (see below). It is possible, that phagocytosis rather than nutrients are responsible for AKT phosphorylation. Due to suggestions ofreviewer 1 we had added experimental results addressing the role of mTORC2 and had to discuss the functional relations between mTORC1/2 and AKT (see below).

 AKT is a major antiapoptotic protein. So why do infected leucine treated CBMO display increased AKT activity and increased apoptosis?

6) The authors conclude: “This leads to the downstream reduction of BCL-2 family  proteins, initiating the intrinsic apoptotic signaling in CBMo”. But no experiment was performed to demonstrate that indeed decreased bcl-2 expression was responsible for apoptosis.

7) In figure 5A and B, BCl2 and BCLxl levels are reduced in infected leucine treated CBMO. This effect is not reversed by rapamycin hence mTORC1 is not responsible for these reduced expressions or bcl2 and bclxl decreased expression are not the cause of apoptosis

Round 2

Reviewer 3 Report

  • Please provide strong recent evidences from the litterature that in 2020 mTORC1 exhibit proapoptotic effects. The idea in 2002 that mTORC1 might regulate Bcl2 and bclxl was not confirmed later.

Publication search revealed an article from Zhang et al. which was published 2014 (doi: 10.1158/1535-7163.MCT-13-0576). In this article the signal transduction axis AKT→mTOR→BCL-2 family → apoptosis is investigated in leucemic tumor cell lines. The authors provide evidence that signalling via Akt phosphorylated at site 308/473 correlates with downregulation of BCL-2 in U937 cells. Blocking of either mTOR and ERK pathway resulted in an increase of apoptosis. We cite this publication in the discussion section

This article demonstrates that combining mTOR and ERK inhibitors exhibits increased proapoptotic effects compared to either treatment alone. Hence this suggests that blocking mTOR increases apoptosis and consequently highlight the anti-apoptotic effect (if any) of mTOR. In addition there is no investigation whether mTOR regulates Bcl2. U937cells  express low levels of Bcl2 which is not related to AKT/mTOR pathway. Therefore other articles are needed to support a proapoptotic role of mTORC1

  • In figure 5A and B, BCl2 and BCLxl levels are reduced in infected leucine treated CBMO. This effect is not reversed by rapamycin hence mTORC1 is not responsible for these reduced expressions or bcl2 and bclxl decreased expression are not the cause of apoptosis

 In the meantime, we conducted more experiments and can provide evidence, that rapamycin does indeed downregulate BCL-2 and BCL-XL in leucine supplemented E.coli infected CBMo. We, accordingly changed the Figs 5 A, B and added new Supplementary Fig3 G-H to provide more evidence, that the reaction is leucine dependent. As well, it has been showed that in PICD the presence of apoptosis markers representing the intrinsic apoptotic pathway (Ying & Häcker, 2020, doi:10.1007/s10495-007-1). We have cited this manuscript in the discussion section (No 48).

Again the conclusions are that leucine increases apoptosis in infected CBMO by increasing mTORC1 activity which is associated with decreased Bcl2 and BclXl expression. Fig 5A and B show however that rapamycin further decrease Bcl2 and Bclxl levels. Hence inhibition of mTORC1 decreases Bcl2 and Bclxl which go against any proapoptotic effects of mTORC1. In this case and to follow the conclusions of the manuscript, we would have expected that rapamycin upregulates Bcl2 and BclXl expression. Altogether this suggests that mTORC1 is not involved in leucine mediated infected CBMO apoptosis

  • Following activation mTORC1 phosphorylates 4E-BP1 and S6K1. In turn S6K1 phosphorylates S6 ribosomal protein. Here, leucine induces mTORC1 phosphorylation in infected CBMO after 4hours (Fig3D) but no increase of 4E-BP1 phosphorylation nor S6 ribosomal protein are detected at 4 hours (Figure 3F and G). Hence, mTORC1 activity is not increased by leucine in infected CBMO unless the authors demonstrate clear increased phosphorylation of another mTORC1 downstream effector at 4 hours

Author Response

This manuscript is a resubmission of an earlier submission. The following is a list of the peer review reports and author responses from that submission.

Round 1

Reviewer 1 Report

Overall revised results are acceptable, but the revised manuscript still has no figures in Figure 3 on page 7.

Reviewer 2 Report

In this manuscript the authors show that leucine increases PICD in infected CBMo presumably by activating mTORC1. While overall the results and their consequences are interesting the major issue is that the authors have failed to correctly demonstrate mTOR signaling pathway in this process. Here are my major concerns:

  • mTOR signaling pathway is barely presented in the introduction. References 8 and 9 are not appropriate to describe mTOR biology. Authors should use some of David Sabatini’s reviews on the topic. Similarly ref 7 is not related to mTOR.
  • Authors state that mTORC1 has proapoptotic effects and cite a 2002 reference (number 10 ). This is not adequate as in 2002 mTORC2 was not even discovered. In fact, very few evidence have demonstrated that mTORC1 induces apoptosis
  • Authors use metformin to increase glucose uptake. However metformin inhibits mTORC1 by activating AMPK. Hence the effect of AMPK(inhibition of leucine induced PTCD) can just rely on mTORC1 inhibition.
  • Authors have to provide the phospho epitopes that were analysed otherwise it is not possible to correctly analyse the results: for instance was it pAKT ser 473 or thr308?
  • Leucine is a major activator of mTORC1 however the authors failed to demonstrate mTORC1 activation following leucine treatment. S6 ribosomal protein is not a direct downstream effector of mTORC1 and gets activated only after 24h. In addition there were no increase in 4EBP-1 phosphorylation.
  • Rapamycin doses and duration of treatment have to be clearly given.
  • Leucine increases pAKT in CBMo infected cells. What is the mechanism behind this effect. Also AKT is a major anti apoptotic signaling intermediary which fully contrasts with the results obtained by the authors.
  • The effect of leucine alone is missing in figure 4. Does the proPICD effect of leucine require E coli infection? In this case how does E coli participate in this mechanism.
  • I do not get the link between reduce glycolysis and increase PICD. In fact leucine does not modify infected CBMo metabolism
  • Authors conclude that the effect on fig 1g could be due to a lack of glucose. What do the mean? It s rather glucose transporters that are different which leads to reduce glucose uptake.
  • S6 should be referred to as S6 ribosomal protein. Also EBP is 4EBP-1
  • The authors did not test the effect of rapamycin on bcl2/ bclxl cytochrome c and cleaved capsase-9. Hence it is not possible to incriminate mTORC1 in this effect.
  • Cartoon in figure 5 needs to be improved. Firstly E coli is not shown however leucine effect seems only to happen in E coli infected cells. Secondly the link between mTORC1 and bcl2 and bclxl has not been demonstrated. Thirdly leucine induces mTORC1 activation in a rheb dependent mechanism as well. Fourthly, what is mTORC2 doing in this cartoon? The link that leucine activates AKT is missing. S6kinase was not investigated at all and should not appear here.

In summary the autors did not convincingly demonstrate the role of mTORC1 in leucine mediated PICD in infected CBMo. In addition most of their results are contrasting with previous well established functions of mTORC1 or AKT. In particular AKT is antiapoptotic but is here activated by leucine. In addition few evidences point to a proapototic role of mTORC1. Therefore a solid discussion should be written to try to explain these discrepanties.

Reviewer 3 Report

The present paper "Leucine induces phagocytosis induced cell death (PICD) in cord blood monocytes by uncoupling glycolysis from mTOR signaling" is complex and present a quite comprehensive study. The topic is interesting and might also be highly clinically relevant.

At this point, as a referee of a revised form of the manuscript, I have mainly focused on minor comments. The manuscript contain several spelling and punctuation mistakes that need to be corrected. The first and most obvious mistake is that the first letter of the abstract is missing. Other examples; line 37 insert space between and and although, period after indicated in figure legend 1, line 430 - antibodies to/against what?, student's t-test should be Student's (line 112, 522). Further, abbreviations should be explained the first time they are used, and figure legends should be complete, including abbreviations. The reader should be able to understand the figures after reading only legend, briefly what has been done, which concentrations, times, what is shown mean with SD or SEM or other, and number of experiments. 

Regarding the results, in figure 1 similar bar charts as in A and B, would have been appreciated in C and D as well. Figure 3 is missing from the file I have access to. The y-axis legend should be explained, i.e. what is ECAR, OCR, LAT-1, MFI and what is the meaning of the numbering of glucose uptake, nmol/mg protein? Similar comments to figure 2; explain the y-axis. Figure 3 is not visible in the file. Figure 5: F, G, H?

The discussion is extensive, and a short conclusion or summary of the main findings at the end would have been appreciated.